# OpenReview forum: "SAFEVISION: Efficient Image Guardrail with Robust Policy Adherence and Explainability"
_ICLR.cc/2026/Conference — Submitted to ICLR 2026_

### Official Review · Reviewer_7qbz · 2025-10-23

**Soundness:** 1
**Presentation:** 2
**Contribution:** 3
**Rating:** 4
**Confidence:** 4

**Summary:**

The paper introduces SafeVision, an image guardrail system designed to enhance the safety and policy adherence of vision-language models (VLMs). The authors propose a training pipeline with a customized loss function, a data collection and generation framework, and the introduction of two datasets: VisionHarm-C and VisionHarm-T. The system aims to be more flexible regarding adaptivity to different safety policies and includes an iterative refinement process for both model weights and safety policy definitions. Experimental results claim improved guardrail accuracy and generalization to new safety categories.

**Strengths:**

* The iterative self-refinement training loop appears to be a novel, interesting approach in the realm of safety.
* The overall structure of the paper is well organized, though some parts miss details or require further clarification (see weaknesses).
* The proposed dataset and training methodology represent a valuable contribution to the visual safety community.
* The paper’s focus on a timely problem of requiring more adaptive vision safeguards.

**Weaknesses:**

### **Methodological Clarity**
* **Line 183 / Fine-tuning Stage** – The initial SigLIP fine-tuning on the “manually collected unsafe dataset” is not explained.
* **Line 203 / Dataset Size** – Include the dataset size of *VisionHarm-T* already when first mentioned.
* **Lines 227-228** – The claim that “Fine-tuning plain VLMs on harmful datasets enables them to serve as guardrail models. However, this straightforward adaptation results in inefficiency and
suboptimal performance.” lacks empirical or cited evidence. Please support this with either an experiment or reference. Generally, I would assume the opposite to be the case as the model is tasked to better understand the
* **Lines 229ff** – “Customizable Guardrail Modes” and “Policy Adherence” are described as *introductions*, though these appear to be inherent capabilities of VLMs in general rather than novel components. Maybe you just improve upon these capabilities.
* **Line 271-272** – Explain why such large efficiency and accuracy improvements (e.g., reduced training/inference time~20%, +1.34 % accuracy) occur. What do you think specifically caused these gains, maybe percentage wise number of output could provide some insides?
* **Line 273 / LLM-based Policy Parser** – How does the LLM-based Policy Parser work?
* **Lines 331-336 / Post-training** – Clarify if this step belongs to the post-training pipeline shown in *Figure 3*. How do DPO, validation data, and GT answer generation relate to Fig.3


### **Dataset Concerns and Safety Policy**
* **VisionHarm-C (Main vs Supplement)** – The main text reports 2 863 samples, while the supplement lists only 401. If this is a subset, please clarify explicitly.
* **VisionHarm-C vs VisionHarm-T** – Why do the category sets differ between both datasets? Justify the decision.
* **Line 203 / Dataset Balance** – *Table 7*: 82 % of *VisionHarm-T* samples fall into only two categories (“Weapons_Substance_Abuse” + “Safe”). This imbalance weakens evaluation reliability.
* **Table 8 / VisionHarm-C** – Although better balanced, some classes (e.g., *Self-Harm* = 9 samples) are too small for meaningful evaluation.
* **Section 3.1 / Category Risk-Guidlines** – Are there explicit risk guidelines or definitions that set the boundary between *safe* and *unsafe* content within your 10 categories? Similar to those of VisionHarm-C (pp. 21–25)
* **Section 3.1 / QA Pairs** – In Figure 6, what is the exact handling when an image falls into multiple safety categories?
* **Chaptrer 3 / Policy Coverage** – A justification on the used safety policy (pp. 21–25) is missing, how was it derived and why does it sufficiently capture the broad notion of “safety.”
* **Chapter 3 / Edge Cases** – How are ambiguous cases like artistic nudity, educational, or historical content handled and classified as safe or unsafe?
* **Chapter 4.3 / Policy Refinement** – Define the difference between *policy v0* and *v1*. Are definitions merely added or also refined? A qualitative comparison would strengthen the paper and provide deep insights about the policy evolves trought the iterations.


### **Figures and Experimental Design**
* **Figure 2** – The bottom-row description is misleading. Please revise to better guide the reader through the figure.
* **Figure 3 (left)** – The diagram is hard to interpret. Clarify what arrows, connectors, and colors (blue/green/orange/red) represent.
* **Figure 4 (bottom row)** – Specify which dataset this part evaluates on. Also justify the use of *AUPRC* given heavy class imbalance. Metrics like balanced ACC or ROC are usually more appropriate for heavily unbalanved class distributions. Describe what threshold you use for AUPRC.
* **Overall Figures** – Some visualizations (e.g., *Figure 3 left*) lack clear semantics and legend details, please make these explicit.

### **Conceptual and Interpretative**
* **Line 325 / “Guardrail Results”** – Please rephrase. It seems you refer to safety ratings or predicted explanations, not the entire guardrail output.
* **line 193-195 / Overfitting Claim** – The claim that fine-tuning “overfit to the guardrail task, rapidly impairing its ability to understand image content, leading to performance drops and loss of policy adherence.” needs explanation. Why would learning to relate image content to policy reduce adherence?
* **Section 5.3 / Baseline Comparison** – clarify what policy inputs baseline VLMs and VLM-Safeguard models receive during inference.
* **Lines 406-410 / Efficiency Comparison** –  claim VLMs are faster than CLIP or CNN models. This is factually incorrect, as VLMs include CLIP-like encoders; please revise or provide evidence.

### **Citation and Reference**

* **Double-check** your *Llama 3* reference for correctness and citation style.

**Questions:**

For suggestions for improvements please see weaknesses

**Details Of Ethics Concerns:**

The work involves the creation, labeling, and distribution of data containing potentially harmful, unsafe, or sensitive visual material (e.g., violence, nudity, self-harm, substance use). Handling such data raises ethical questions regarding data sourcing, annotator exposure, consent, content moderation, and safe redistribution. I would recommend an ethics check to ensure:

* All data were collected and labeled in compliance with ethical and legal standards (e.g., copyright, privacy, and consent).
* Appropriate usage restrictions and access controls are in place for sharing the VisionHarm datasets and the SafeWatch model

---

### Official Review · Reviewer_Mqxr · 2025-10-26

**Soundness:** 2
**Presentation:** 3
**Contribution:** 2
**Rating:** 2
**Confidence:** 3

**Summary:**

This paper introduces a high-quality dataset, VISIONHARM, and proposes an image guardrail model, SAFEVISION, to address the challenge of unsafe visual content. VISIONHARM consists of two subsets covering diverse harmful categories, while SAFEVISION integrates an effective data collection and generation framework, a policy-following training pipeline, and a customized loss function. The authors conduct extensive evaluations on several existing benchmarks and demonstrate strong performance.

**Strengths:**

- New dataset contribution.
- New image guardrail.
- Strong performance.

**Weaknesses:**

- The Related Work section (Section 2) lacks sufficient discussion of prior studies, especially detailed comparisons with state-of-the-art works [a,b]. For instance, I found two recent papers whose dataset construction and image guardrail designs are highly similar to the authors’ contributions. From the data collection perspective, both works incorporate multiple policies. LlamaGuard also claims to support flexible policy configurations, and UnsafeBench collected a large number of unsafe images from the LAION dataset. Moreover, both works propose new image guardrails. The authors should elaborate more clearly on the distinctions between their work and these prior studies to highlight their novel contributions.

- Although I greatly appreciate the authors’ effort in building the dataset, several aspects remain ambiguous and require further clarification. For example, VISIONHARM covers multiple harmful categories, why were these specific categories chosen? In constructing VISIONHARM-C, manual annotation was used, yet the annotators’ expertise and background, annotation guidelines, and inter-annotator agreement were not clearly described.

- The authors adopt a VLM-based consistency filter to refine the dataset, but its accuracy is not reported. Are there potential false positives or false negatives? In addition, the paper employs an LLM-as-a-judge evaluation metric, but there is no supporting evidence to show that this metric is reliable or capable of producing accurate assessments.

[a] LLAVAGUARD: An Open VLM-based Framework for Safeguarding Vision Datasets and Models
[b] UnsafeBench: Benchmarking Image Safety Classifiers on Real-World and AI-Generated Images

**Questions:**

- What are the differences between the proposed approach and related work?
- Can you provide more details about the data collection process?
- Is the VLM-based filter and LLM-as-a-judge reliable in your evaluation?

---

### Official Review · Reviewer_tjVV · 2025-10-29

**Soundness:** 2
**Presentation:** 1
**Contribution:** 2
**Rating:** 2
**Confidence:** 4

**Summary:**

This paper proposes SAFEVISION, an image guardrail system built on top of InternVL-2.5-2B/8B. The author claims three main contributions: 1. A dual-mode (classification and comprehension) pipeline for fast inference and explainability. 2. self-refinement training loop involving policy updates and weighted loss. 3. A new dataset, VISIONHARM-T/-C, for harmful-image detection. The evaluation shows that the proposed method outperforms GPT-4o and other baselines in accuracy and inference speed across their own benchmarks, UnsafeBench, and LLaVAGuard Dataset.

**Strengths:**

1. This paper introduces a newly curated datasets that could support future safety research.
2. The authors explicitly consider computational efficiency, which is a crucial factor when deploying safety guardrails.

**Weaknesses:**

My concerns fall into two main categories — experimental soundness and technical novelty.

1. SAFEVISION is compared to safety baselines (e.g., Q16, Multi-Headed, LLaVAGuard) that could also be fine-tuned on the proposed dataset, but appear not to be. In addition, SAFEVISION’s refinement process uses powerful external models (e.g., GPT-4o, Qwen-VL, InternVL) unavailable to baselines, making results not directly comparable. The authors are encouraged to retrain baselines under comparable conditions.

2. The paper does not report how much each component—especially the post-training and DPO stages—contributes to the overall performance. An ablation study quantifying individual effects would make the results more convincing.

3. Although the paper emphasizes adaptability to unseen categories, Figure 4 shows that vanilla VLMs achieve comparable performance, implying that zero-shot behavior mainly comes from the base model’s pretraining rather than the proposed refinement. In addition, several of the “novel” categories appear closely related to concepts already present in the pretraining data, which further advantages SAFEVISION over baselines that were not pretrained on their datasets.

4. The “refinement process” for safety alignment is largely similar to prior work [1] and lacks clear methodological innovation. Clarifying how this framework differs—either in algorithmic design or policy-update mechanism—would improve the paper’s contribution.

5. The paper’s presentation could be substantially improved. For instance, citations are incorrectly formatted, and key experimental results would be clearer if presented in well-structured tables.

[1] Customize Multi-modal RAI Guardrails with Precedent-based predictions, COLM 2025.

**Questions:**

Please see my comments above.

---

### Official Review · Reviewer_4yYE · 2025-10-29

**Soundness:** 2
**Presentation:** 2
**Contribution:** 2
**Rating:** 2
**Confidence:** 3

**Summary:**

This paper proposed SAFEVISION, an image guardrail model that integrates human-like reasoning to enhance adaptability and
transparency. They also introduced the VISIONHARM dataset that spans diversified harmful categories. Through various experiments, they demonstrated the effectiveness of the SAFEVISION across different benchmarks in image content moderation.

**Strengths:**

1. The paper is in good structure; easy to follow
2. The idea of building image guardrail models is important to the community.

**Weaknesses:**

- The motivation for introducing the VISIONHARM is vague to me. What is the difference between your proposed benchmark and the safety dataset in the VLM domain (e.g., [1] since they also contain many unsafe images)
- I am concerned about the performance of the baselines such as GPT-4o and Llama-guard; They perform poorly on the Weapon dataset. Have you tried to tune their prompts (e.g., by giving more information about what is unsafe in the context?) Then, how do you explain their poor performance?
- Why do you use these 10/15 categories as the unsafe category? Are there any advantages over other benchmarks by using this taxonomy?
- Normally, for aligned VLMs, they would generate rejected responses like ‘Sorry, I cannot’ when input with unsafe images. Then, why would these models be able to generate meaningful QA pairs in your data pipeline?
- The pipeline is like a combination of traditional training strategies without any insights grounded in experiments or proof. Although a per-token loss is introduced, this is actually not so novel since similar ideas have been explored in previous papers (e.g., [2]). To be honest, a method without too many "novel" designs is okay if it solves the real challenges, but I would expect the authors to justify why the existing SOTA models (e.g., GPT-4o and Llama-guard) miss the attributes you are targeting: reasoning and explainability, as they can also generate human-readable reasoning.

[1] Lee Y, Kim K, Park K, Jung I, Jang S, Lee S, Lee YJ, Hwang SJ. HoliSafe: Holistic Safety Benchmarking and Modeling with Safety Meta Token for Vision-Language Model. arXiv preprint arXiv:2506.04704. 2025 Jun 5.
[2] Zeng Y, Liu G, Ma W, Yang N, Zhang H, Wang J. Token-level direct preference optimization. arXiv preprint arXiv:2404.11999. 2024 Apr 18.

**Questions:**

> "tokens related to guardrail results are more critical than those related to image content."
Any supportive evidence for this argument?

---

### Meta-Review · Area_Chair_LbzW · 2026-01-01

**Summary:**

The paper studies important problems relating to the safety of CV/AI systems dealing with images containing potentially harmful content.

The study presented involves:
- various analyses of leading VLM models including InternVL2_5, LLaVaGuard, GPT-40, LLamaGuard3, as well as classification models including Q16, Azure API and several "X-detection" models
- curation of a new dataset of harmful images spanning several categories (VisionHarm)
- a training recipe delivering an improved model, according to the evaluations presented

While I personally acknowledge the significant effort by the authors to put forward such an extensive study, reviewers shared several concerns regarding data collection, experiment setup, and technical novelty.

Given the sensitivity of the topic, and the extra care needed to conduct and report the analyses involved, it warrants holding off on publication till those concerns are resolved.

**Reviewer Concerns:**

Unfortunately, no rebuttal was submitted for any of the reviews.

**Reviewer Scores:**

Reviewers largely share key concerns, regarding data collection, experiment setup, and technical novelty.
This is also reflected in the score distribution 2/2/2/4, strongly suggesting a final score in the same range.

---

### Decision · Program_Chairs · 2026-01-26

Reject